# Molecular Dynamic Simulations and Experiments Study on the Mechanical Properties of HTPE Binders

**DOI:** 10.3390/polym14245491

**Published:** 2022-12-15

**Authors:** La Shi, Xiaolong Fu, Yang Li, Shuxin Wu, Saiqin Meng, Jiangning Wang

**Affiliations:** Xi’an Modern Chemistry Research Institute, Xi’an 710065, China

**Keywords:** HTPE binders, molecular dynamic simulation, mechanical properties, crosslinking structures

## Abstract

The mechanical properties of HTPE binders have been systemically studied through combining the microstructure molecular simulations with macroscopic experiments. In this study, the crosslinking structures of HTPE binders were established by a computational procedure. Based on the optimized crosslinking models, the mechanical properties and the glass transition temperatures (T_g_) of HTPE/N-100, HTPE/HDI, HTPE/TDI, and HTPE/IPDI binder systems were simulated; specifically, the T_g_ were 245.758 K, 244.573 K, 254.877 K, and 240.588 K, respectively. Then the bond-length distributions, conformation properties, cohesive energy densities, and fraction free volume were investigated to analyze how the microstructures of the crosslinking models influenced the mechanical properties of HTPE binders. Simultaneously, FTIR-ATR spectra analysis of HTPE binders proved that the special peaks, such as -NH and -NCO, could be seen in the crosslinking polyurethane structures synthesized between prepolymers and curing agents. The dynamic mechanical analysis was carried out, and it found that the T_g_ of HTPE/N-100, HTPE/HDI, HTPE/TDI, and HTPE/IPDI binder systems were −68.18 °C, −68.63 °C, −65.67 °C, and −68.66 °C, respectively. In addition, the uniaxial tension verified that both the ultimate stress and Young’s modulus of HTPE binder systems declined with the rising temperatures, while the strains at break presented a fluctuant variation. When it was closer to glass temperatures, especially −40 °C, the mechanical properties of HTPE binders were more prominent. The morphology of the fractured surface revealed that the failure modes of HTPE binders were mainly intermolecular slipping and molecular chain breakage. In a word, the experimental results were prospectively satisfied using the simulations, which confirmed the accuracy of the crosslinking models between prepolymers and curing agents. This study could provide a scientific option for the HTPE binder systems and guide the design of polyurethanes for composite solid propellant applications.

## 1. Introduction

Compared with double-based propellants, such as modified double based propellants and other composite solid rocket propellants, hydroxyl-terminated polyether (HTPE) propellants performed better in insensitive munition (IM) properties [1,2,3,4] and had more favorable mechanical properties over a wide temperature range [5,6,7]. HTPE propellants consisted of solid particulate oxidants, metal fuel, some assistant reagents, plasticizers, and HTPE binders, which were block copolymers synthesized by tetrahydrofuran and ethylene glycol [8,9]. HTPE binders would form polyurethane three-dimensional networks when the prepolymer reacted with the curing agents. The linear prepolymer terminated with -OH groups; the curing agents provided -NCO [10,11,12] groups. While the solid particles were combined together, the desired mechanical properties would be given to HTPE propellants [13]. Therefore, the mechanical properties of HTPE propellants mainly depend on the structures and compatibility of binders [14,15,16,17]. The mechanical and thermal properties of HTPE binders would be mainly contributed by the dangling chains (chains folding due to the various CNO/OH ratios), the crosslink average molecular weight [5], and the crosslink density [18]. The curing reaction kinetics of the HTPE polymer with hexamethylene diisocyanate biuret were studied by simultaneous rheometry and FTIR spectroscopy, from which the fluidity, elastic modulus, and characteristic intensity peak could be observed [19]. Shen et al. [20] controlled the mass contents of the poly(ε-caprolactone) to the HTPE polymer and investigated the mechanical and thermal properties of the HTPE/poly(ε-caprolactone) propellant from −50 °C to 70 °C. Wen et al. [21] added different proportions of a novel hyperbranched multi-arm azide copolyether to the HTPE polymers and observed perfect thermal stability. The modified hyperbranched polyesters were introduced into the HTPE propellant [22], leading to a reinforced creep resistance and relaxation property. Konrad et al. [23] simulated the bond formation and dissociation processes of epoxy resin, which obtained high curing degree structures and explained the competition of the bond formation and dissociation. The relationship between HTPE polymers and the different curing agents was contrastive and analytical [24]. The curing process and mechanical properties of toluene diisocyanate (TDI) and polyfunctional isocyanate (N-100) curing systems were investigated [25]. The synthesis and characterization of HTPE binders were carried out [26,27,28] and compared with the Hydroxy Terminated Polybutadiene (HTPB) binders [29]. The mechanical properties of the solid propellant would influence its storage, burning, explosion, and so on. Scientists have performed many methods to study the HTPE propellants, such as high-strain-rate mechanical response under a modified split Hopkinson pressure bar [30], the HTPE polymers with the mixed N-100 and isophorone diisocyanate isocyanate (IPDI) by in situ-preparation [31], one-dimensional time to explosion to validate chemical mechanisms [32], the minimum free energy method, and the miscibility of polymers and plasticizers [33]. However, the systematic mechanism of mechanical properties is still deficient, and it is still essential to combine the simulations with experiments to explore the mechanical properties of HTPE binders, which could explain the macro-properties through the micro-structures and really predict the mechanical and thermal properties [34].

In this paper, molecular dynamic simulations and experiments were applied to investigate the mechanical properties of HTPE binder systems. On the one hand, the various degrees of crosslinking HTPE models were constructed by Perl scripts, which were accomplished with the set cut-off distances under the repeated molecular dynamic simulations. Then the crosslinking models were used to analyze the mechanical properties and the glass transition temperatures (T_g_) of HTPE binders. Furthermore, bond-length distributions, conformation analysis, cohesive energy density, and friction-free volume were evaluated to verify how the microstructures influenced the mechanical properties. On the other hand, all HTPE binder system experiments were performed, including FTIR-ATR spectra analysis, dynamic mechanical analysis, uniaxial tensile, and morphology of the fractured surface.

## 2. Simulations

### 2.1. The Model Constructions and Molecular Dynamic Simulations

All molecular models were built by Materials Studio; construction information is found in the literature [35,36]. The structures of hydroxy-terminated polyether (HTPE), polyfunctional isocyanate (N-100), hexamethylene diisocyanate (HDI), toluene diisocyanate (TDI), isophorone diisocyanate (IPDI), and the four blends are shown in Figure 1. The HTPE chains were built with repeat ethylene glycol and tetrahydrofuran in the Builder Polymers module, whose average molar weight was 3974 g/mol.

The blends were constructed in the Amorphous Cell module with the box of 90.39 Å × 89.46 Å × 90.48 Å and the initial density of 0.6 g/cm^3^. Then 10 frames were output; the one with the lowest number was expected to be the basic structure. The van der Waals and electrostatic interactions selected the atom-based and Ewald methods. Meanwhile, the simulation quality was fine.

The molecular force field was the core of molecular mechanics, which calculated the equilibrium structure and energy of molecules based on the classical Newtonian equations of mechanics. The bulk properties of HTPE binder models could be extracted if a good approximation of the potential in which atomic nuclei move was available and if there were methods that could generate a set of system configurations that were statistically consistent with a full quantum mechanical description. After the molecular models were built, the geometric structures were relaxed with 50,000 steps of energy minimization. Simulations started by choosing an initial state by setting the initial positions and velocities of all the particles. The initial velocities were usually set by choosing each component of the velocity vector of each particle from the Maxwell distribution. HTPE/N-100, HTPE/HDI, HTPE/TDI, and HTPE/IPDI binder models were simplified as S-1, S-2, S-3, and S-4, respectively. Then, the amorphous blends were annealed from 600 K to 300 K with an interval temperature of 20 K and 10 annealing recycles in total at each temperature, which contained 200 picosecond (ps) dynamics performing every temperature ramp in the NPT ensemble, NHL thermostat, and Berendsen [37] barostat. Furthermore, the time step was 1.0 femtosecond, which was chosen to be small as compared to the shortest fundamental time scale in the Hamiltonian, while not being so small as to limit the efficiency of the program. The MD time averages were equivalent to the averages over the microcanonical ensemble. These models need further dynamics to equilibrate the system with the NPT [38] ensemble for 400 ps. In this paper, all models were optimized and analyzed under CompassII force fields [39] and forcefield-assigned charges.

### 2.2. The Curing Crosslinked Procedure

Making the curing crosslinked structures similar to the experimental structures has attracted many scientists [23,40]. The prepolymer reacts with the curing agents under the principle of addition polymerization. For most polyurethane, the curing reaction mechanism of HTPE binders is as shown in Figure 2b. When the annealed models were all optimized thoroughly, the mixed prepolymer and curing agents’ molecular chains would execute a crosslink procedure to attain polyurethane structures.

The flow chart of the crosslinking reaction mechanism algorithm, as seen in Figure 2a, was accomplished automatically by Perl scripts through changing the cutoff distances between the active atoms of -OH groups and the active atoms of -NCO groups. Furthermore, some assumptions were premised. The equal chances of HTPE, N-100, HDI, TDI, and IPDI molecular chains were given to produce crosslinking structures. The -O-H and -N=C=O bonds broke and formed among reactive atoms of -NCO and -OH groups when they met in the specified distance range. This algorithm would finish when it reached the maximum cutoff distance or reached the set conversation. Rather than putting the whole polyurethane into boxes directly, the random crosslinked structures based on atomic collision can really reflect the experimental samples. The reactive distance would increase from 3.5 Å to 14.5 Å with a ramp of 0.5 Å. The maximum conversion (α), system temperature (T), minimum reaction radius (R_min_), maximum reaction radius (R_min_), and other parameters could be reset flexibly.

When the crosslinking models were established, it was essential to make energy minimizations and dynamic optimizations for these models. In order to attain utterly relaxed crosslinking structures, one more annealing producer needed to be practiced again. These operation progresses could refer to the origin blended models building. Up to now, the basic crosslinking models were built successfully, which would be used to compute the special properties of HTPE binders. The final curing conversations of HTPE/N-100, HTPE/HDI, HTPE/TDI, and HTPE/IPDI were 95%, 85%, 100%, and 80%, respectively, whose crosslinking models can be seen in Figure 2c.

## 3. Experimental Section

### 3.1. Materials

Herein, the main prepolymer, curing agents, and some essential additives are listed. Hydroxy-terminated polyether (HTPE, #61) was bought from Liming Research & Design Institute of Chemistry Co., Ltd. (Luoyang, China), whose average molecular weight was 3974 g/mol, the polydispersity was 1.58, and the hydroxyl value was 4.729 mol/g × 10^4^. Polyfunctional isocyanate (N-100) was provided by the Xi’an modern chemistry research institute with 5.32 mmol/g of isocyanate group. Toluene diisocyanate (TDI; AR) was provided by Sinopharm Chemical Reagent Co., Ltd., (Shanghai, China). Hexamethylene diisocyanate (HDI; 99% purity) was bought from the Aladdin factory. Isophorone diisocyanate (IPDI; 99% purity) was provided by Shanghai Macklin Biochemical Co., Ltd., (Shanghai, China). The ALT-402, as a water remover, was provided by Changde Ailite New Material Technology Co., Ltd., (Zaozhuang, China). Tripheny bismuth (TPB), as a catalyst, was bought from Tanyun Chemical Research Institute Co., Ltd, (Liaoning, China). After the four systems had been cured and shaped in the constant temperature oven, the binders were cut into the standard size for chemical structure analysis, thermomechanical property tests, and fractured morphology characteristics.

### 3.2. Preparation of HTPE Binders

Firstly, the prepolymer HTPE melted from a white lax state to a uniform, clear, and transparent liquid at 85 °C for at least 5 h. Secondly. 2wt%~3wt% ALT-402 were added into the pure HTPE solution, and the blends were mixed well, and then dried with extra water in a vacuum oven at 60 °C for 72 h. Thirdly, the curing agents N-100, HDI, TDI, and IPDI were instilled with the NCO/OH group molar ratio amounts of 1.2, respectively. The TPB as a catalyst was added at 0.3wt%. Fourth, all raw materials were mixed for 10 min to disperse uniformly and the compounds were degassed for 30 min in the vacuum oven. Finally, the blends were slowly poured into two different molds and were put into a water bath incubator for more than 5 days at 70 °C until the curing reactions were finished completely. A simplified experimental route to manufacture HTPE binders is shown in Figure 3a. The final experimental samples of HTPE binders can be seen in Figure 3b.

### 3.3. Apparatus and Characterizations

The dynamic mechanical analyzer (DMA850, TA, USA) could study the dynamic mechanical properties of the S-1, S-2, S-3, and S-4 binders with the tensile clamp. The HTPE binder specimen size was 35 mm × 13 mm × 3 mm. The scanning temperature range was −120 °C to 0 °C and the low temperature was controlled with liquid nitrogen. The system heating rate was 3 °C/min and the loading frequency was 1 Hz. The final result was analyzed from three samples under the same test conditions.

When the binders had been cut into JANNAF dog bones, the mechanical properties, such as Young’s modulus, the ultimate tensile stress, and the strain at break, could be tested at −40 °C, 20 °C, and 50 °C and the loading rate was 500 mm/min. Five specimens (length 75 mm × narrow parallel width 4 mm × thickness 2 mm) of each binder system were needed, and the average values were determined, which could diminish the accidental error. Moreover, the universal testing machine (AG-X plus, 5kN, Shimadzu, Kyoto, Japan) was equipped with an incubator, which could guarantee the sample testing at special temperatures. The details could be referred to in the standard GB_T528_2005.

Fourier transformed infrared-attenuated total refraction (FTIR-ATR, TENSOR #27, Bruker, Bremen, Germany) was used to analyze the chemical structures of HTPE binders. Its optimum resolution was 0.5 cm^−1^ and provided a wavenumber range of 400~4000 cm^−1^.

Scanning Electron Microscope (SEM; Quanta 600F, FEI, Hillsboro, OR, USA) presented a magnification for the breakage surface morphological inspections of the HTPE binders. The accelerating voltage was 20 kV, the working distance was 10.2 mm, and all microcrack surfaces were coated with conductive gold films.

## 4. Results and Discussions

### 4.1. The Simulation Section

#### 4.1.1. Glass Transition Temperature

The amorphous polymers could evolve from a malleable liquid or rubbery state to a glassy state at the glass transition temperature (T_g_) [41,42,43,44]. With a low T_g_ for the polymers, especially, nitrocellulose, polyethylene glycol, hydroxy terminated polybutadiene, glycidyl azide polymer, ethylene oxide-tetrahydrofuran co-polyether, and so on, the energetic materials would reach the ideal physical properties by adjusting the varieties of the prepolymers, the curing agents, and adding the plasticizer. The glass transition temperature directly determined the process and application temperature range of the thermoset polymers, which was an important property of materials and could help realize the mechanical and other thermal properties. Therefore, it is meaningful to explain the macroscopical transition phenomenon regarding the T_g_ in the molecular view. Additionally, analyzing the function of mean-squared displacement of molecules [45], density change, expansion and contraction of volume, expansion coefficient, and the total energy to temperature could estimate an accurate T_g_. Furthermore, the simulated T_g_ could guide the experimental designs effectively.

The optimized crosslinking models were cooled from 450 K to 150 K with a 10 K interval. 50 ps NPT dynamic simulations were practiced in every temperature ramp and the final densities were averaged from five frames. Furthermore, all the data were computed and extracted by a Perl script. The T_g_ was the point of abscissa intersection, which was derived from two linear lines fitting to the densities at low temperatures and high temperatures. Figure 4 shows the density-temperature curves of HTPE binders. The T_g_ values of the S-1, S-2, S-3, and S-4 binder systems were 252.567 K, 244.573 K, 257.877 K, and 240.588 K, respectively. All the binder models showed an approximate T_g_ as a result of a few different curing agents existing in the crosslinking polyurethane structures. This result confirmed the vital effect of HTPE polymer structures on the mechanical properties and the connection roles of the curing agent.

#### 4.1.2. Mechanical Properties

In this paper, the static constant strain method was used to analyze the mechanical properties of four HTPE binder systems, and the number of strains was six with a maximum strain of 0.003. Of course, all basic models need to be preoptimized structures. Through the built-in procedures, the elastic stiffness constant’s matrix would be output. The stress and strain obeyed Hooke’s law, as shown in Equation (1), and were calculated using the elastic constant.
(1)$$\sigma_{\mathrm{ij}}{=C}_{\mathrm{ij}}\varepsilon_{\mathrm{ij}}\;$$
where σ_ij_ (i, j = 1, 2, 3) was the stress and ε_ij_ was the strain. C_ij_ was the elastic constant, which could be inferred from the [C_ij_] stiffness matrix.

HTPE binders are isotropic polymers, and the Lame’s constants, λ and μ, could be deduced from Formulas (2) and (3).
(2)$$\lambda =\frac{1}{6}{(C}_{12}{+C}_{13}{+C}_{21}{+C}_{23}{+C}_{31}{+C}_{32})$$
(3)$$\mu =\frac{1}{3}{(C}_{44}{+C}_{55}{+C}_{66})$$

According to the basic constants calculated above, the mechanical properties, such as Young’s modulus (E), shear modulus (G), bulk modulus (K), and Poisson’s ratio (υ), would be further concluded in Equations (4)–(7).
(4)$$E=\frac{\mu (3\lambda +2\mu )}{\lambda +\mu }$$
(5)G=μ
(6)$$K=\lambda +\frac{2}{3}\mu$$
(7)$$\nu =\frac{1}{2(\lambda +\mu )}$$

The final mechanical property results of S-1, S-2, S-3, and S-4 binders are listed in Table 1. From Table 1, the S-2 binder system showed the most desired mechanical properties, whose G, K, and E were 2.021 GPa, 6.285 GPa, and 5.477 GPa, respectively. The S-1 and S-3 binder systems presented close G and E values. While S-4 binders presented the least shear modulus, 1.793 GPa, and Young’s modulus, 4.909 GPa, their bulk moduli were higher than the S-1 and S-3 binder systems. As the constant strain simulation method was carried out at 273.15 K, in which equilibrium system structures were minimized after each step; few differences appeared between simulated data and experimental ones. In the HTPE binder-simulated systems, only the different crosslinking conversions and the different curing agents were so, that it was obvious what influence the mechanical properties of HTPE binder systems caused. In a word, these results were in good agreement with the glass transition temperature analysis above.

#### 4.1.3. Bond-Length Distribution

Bond-length distributions of the final crosslinking structures, prepolymer, N-100, HDI, TDI, and IPDI were measured, as seen in Figure 5. Compared to the original prepolymer and curing agents, the new group, -NH, emerged in all crosslinking structures; it declared that the crosslink scripts ran successfully as expected. Especially for the S-3 binder model, -OH groups no longer existed because the crosslinking conversation was 100%. Nevertheless, other partial crosslinked models still contained a few -OH groups. -C(O)O- groups of the crosslinked models were in a wider range than -C-O- groups of prepolymers. The curing agents were excessive, as the curing ratio was 1.5, so that -N=C- groups were still existing in the four final crosslinked models. When it came to different crosslinked models, the original bond-length distributions of the same positions might show a few movements at a specific distance and probability density. It was proven that the curing reaction had an insightful influence on the molecular configurations, even if they were not at the active sites.

#### 4.1.4. Conformation Properties

For the polymers, the lengths and conformations of molecular chains had an insignificant effect on their flexibility, which would directly determine their mechanical properties. The more flexible the molecular chains were, the lower the T_g_ of the HTPE binders. The curing agents worked as links to the crosslinking structures. The HTPE prepolymers were composed of tetrahydrofuran and ethylene glycol, so that there were enough steric hindrances to the flexibility of the final HTPE binders. The high synergy rotational energy barrier would restrain the movements of the molecular chain segments, which could cause worse flexibility.

Two specific points of torsions were screened to calculate the synergetic rotational energy barrier to analyze the conformation behaviors. The torsion of rotation of two specific bonds, Φ1 and Φ2, in the S-1, S-2, S-3, and S-4 binder systems are seen in Figure 6. In addition, the total synergetic rotational energies of the two special bonds were the functions of the torsion angles shown in Figure 7. Compared to other HTPE binders, the S-3 binder system reached the highest synergetic rotational energy, 7979.79 kcal∙mol^−1^_,_ but the S-4 binder system was in the lowest one, 6892.81 kcal∙mol^−1^. The S-1 and S-2 binders showed close values of 7375.45 and 7390.12 kcal∙mol^−1^. As a result, the flexibility decreased in the order of S-4, S-1, S-2, and S-3. The conformation analysis kept the same simulated glass transition temperature.

#### 4.1.5. Cohesive Energy Density

The cohesive energy density (CED) of polymers was relevant to the intermolecular chains, so it could reflect the thermal and mechanical properties of materials. The smaller the CED was, the lower the T_g_ and the higher the elastic moduli were [46,47]. The cohesive energy (E_coh_) was defined as the average energy required when all molecules were removed to a distance of infinity from each other; it can be calculated using Equation (8).
(8)$$E_{\mathrm{coh}}=\langle E_{\mathrm{inter}}\rangle =\langle E_{\mathrm{total}}\rangle -\langle E_{\mathrm{intra}}\rangle$$
where E_total_ is the whole energy of the polymer systems, E _inter_ is the whole energy among all molecular chains, and E_intra_ is the intramolecular energy.

The cohesive energy density is the average cohesive energy per unit volume and is calculated using Equation (9).
(9)$$\mathrm{CED}=\frac{E_{\mathrm{coh}}}{V}$$
where E_coh_ is the internal energy and V is the volume of the polymer system.

The CED of the HTPE binder systems is displayed in Table 2. It was easily found that the CED of the S-4 binder system was 3.062 × 10^8^ J/m^3^, which was the lowest one among the four binder systems. The S-1 binder system had a maximum value of 4.579 × 10^8^ J/m^3^. It indicated that the S-4 binder system showed the lowest T_g_. Of course, these results gave the verifications to the front simulated glass transition temperature and mechanical property analysis.

#### 4.1.6. Fraction Free Volume

The mechanical and thermal properties of the polymers were prominently influenced by the free volumes and movements of molecular chain segments. Investigating the fraction free volume at a molecular scale would help us perceive how the atoms accumulated and the spatial space existed. The lower free volumes of the polymers would lead to a higher T_g_ and elastic modulus. According to the free volume theory, the volume (V_T_) of liquid or solid substances is composed of two parts: the van der Waals volume occupied by molecules (V_0_) and the other one is the free volume (V_f_), the interspace among the molecules. In view of the different volumes of various polymer systems, it is necessary to analyze the FFV of different curing crosslinked polyurethane systems. The distributions of V_0_ and V_f_ are displayed in Figure 8. The concrete information is shown in Table 3. Meanwhile, the FFV was expressed in Equation (10), and it was calculated by the connolly surface in the atom volume and surface modules.
(10)$$\mathrm{FFV}=\frac{V_{f}}{V_{0}{+V}_{f}}\times 100\%$$

The FFV values of the S-1, S-2, S-3, and S-4 binder systems were 22.0%, 17.9%, 17.9%, and 18.0%, respectively. It was found that the FFV of the S-1 binder system was the highest among the four systems, and its crosslink structures were the most compact, which was because of the special structures of the N-100 curing agent. For the other three binder systems, the FFV was nearly 17.9%, and it was collectively affected by the various curing agents, the degree of crosslinking, and the final polyurethane structures. Therefore, it was predicted that the T_g_ and the modulus of the S-1 binder system would be the highest among the four HTPE binder systems. In summary, the fraction free-volume analysis was consistent with the front simulated results.

### 4.2. The Experiment Section

#### 4.2.1. FTIR-ATR Analysis

FTIR-ATR spectroscopy was widely used to indicate the structures of the solid propellants, especially the curing reactions and the polyurethane binders. The main functional groups of the HTPE binders were studied by the FTIR-ATR technique, and the infrared spectra of the four binder systems are shown in Figure 9. A wide absorption band of amide -N-H groups stretching vibrations at 3326 cm^−1^ could be seen in all specimens. The -C-H groups of the HTPE binders, prepolymers, and curing agents presented the stretching vibration peaks at 2939 cm^−1^ and 2852 cm^−1^. The absorption peak at 2260–2280 cm^−1^, which was attributed to the asymmetric stretching vibration peak of -NCO, performed a large absorption intensity. Therefore, it indicated the most effective characteristic peak for identification of -NCO groups, which proved that the carbamate structures had been created between the prepolymers and curing agents. The symbol peaks of polyurethane could be observed at 1690 cm^−1^ and 1735 cm^−1^. The stretching vibration peaks of C-O bonds in the ester group were at 1230–1220 cm^−1^, and the stretching vibration peaks of C-O bonds in the ether group were at 1115–1080 cm^−1^. However, a few differences among the various HTPE binders were also exhibited, such as the intensity of the main peaks, the position of the peaks, and even the numbers of the peaks. These behaviors were mainly caused by the various kinds of curing agents, the extra curing agents, and the final polyurethane structures. In a word, the new urethane groups could be significantly noticed in the FTIR-ATR spectra curves, and the structures of HTPE binders were successfully crosslinked by the experimental methods.

#### 4.2.2. Dynamic Mechanical Analysis

The mechanical properties of the HTPE propellants were mainly determined by the prepolymer, but the effect of different curing agents was also worthy of exploration. The dynamic mechanical analyzer (DMA) machine could test the viscoelastic behavior of the HTPE binders, and it could match the glass transition temperature more accurately than the differential scanning calorimetry methods. Figure 10 shows the curves for the storage modulus (E’), loss modulus (E”), and loss factor (tanδ) as the functions of temperatures when the loading frequency is a constant. From Figure 10, the mechanical properties of all the HTPE binder systems showed similar trends with the increasing temperatures. E’ decreased rapidly in the low temperatures and tended to level off in the high temperatures. The main reason was that the interactions of the molecular chains were stronger in the low temperatures and the forces of the intermolecular were weaken when the molecules gained more freedom in the high temperatures. At the initial temperature, the E’ of the S-1 and S-3 binder systems were higher than the others. The loss modulus could measure how much the energy was converted into heat during the viscosity deformation stage. Although the loss modulus curves of the S-1 binder system were similar to those of the S-3 binder system, the peak value of the S-1 binder system was higher than that of the S-3 binder system. The S-2 binder system had the lowest peak value of E” under the experimental temperatures. The crosslinking conversion rate and the kinds of curing agents made the final crosslinking structures different, which caused the various flexibility of the intermolecules, and varying thermodynamic mechanical properties of the HTPE binders.

The loss factor was defined as the ratio of the loss modulus to the storage modulus, and the peak abscissa of the tan δ curves was the glass transition temperature (T_g_) [48]. The tan δ curves of the HTPE binder systems are seen in Figure 10. We could conclude that the T_g_ values of the S-1, S-2, S-3, and S-4 binder systems of the HTPE binder systems were −68.18 °C, −68.63 °C, −65.67 °C, and −68.66 °C, respectively. The lower the T_g_ was, the flexibility of the intermolecular was more favorable. In a word, the experimental T_g_ fitted the trend of the molecular dynamic results but were slightly lower than the simulated ones, which were owed to the cooling rates of the molecular dynamic simulation method differing from the rising rates of the dynamic mechanical analyzer [49].

#### 4.2.3. Uniaxial Tensile Analysis

It is a traditional way to study the mechanical properties of solid propellants with an uniaxial tensile machine. The strain-stress curves of the S-1, S-2, S-3, and S-4 binder systems are exhibited in Figure 11. At −40 °C, 20 °C, and 50 °C, the ultimate tensile strength (σ_m_), strain at break (ε_b_), and Young’s modulus (E_exp_) are summarized in Table 4, which could be calculated according to Equations (11)–(13).

From Figure 11, the S-1 binder system showed the hackly curves at 20 °C and 50 °C while the other binder systems had the smoother ones, which may be due to the multi-functional isocyanate structures. The four systems showed that the strain-stress curves were concave, similar to an exponential function, at 40 °C and shifted into the logarithmic function at 20 °C and 50 °C. The HTPE binders were at low glass transition temperatures; the interaction force of molecular chains was more powerful at 40 °C.
(11)$$\sigma_{m}=\frac{F}{A}$$
(12)$$\varepsilon_{b}=\frac{∆L}{L}$$
(13)$$E_{\exp}=\frac{\sigma_{m}}{\varepsilon_{b}}$$

The σ_m_, ε_b_, and E_exp_ of the four binder systems gradually decreased with increasing temperatures, and they reached their maximum values at −40 °C. Compared with the other three binder systems, the S-2 binder system had the highest ultimate strength value of 29.057 ± 0.559 MPa at low temperatures. At the same temperature, the S-1, S-3, and S-4 binder systems realized their own maxima of 8.356 ± 1.949, 13.405 ± 2.001 and 24.435 ± 3.841 MPa, respectively. When it came to the room temperature and high temperature, all the σ_m_ attained relatively smaller values. The σ_m_ of the S-2, S-3, and S-4 binder systems stayed approximately at 2.0 MPa, but the S-1 binder system was less than 0.5 MPa. Meanwhile, the HTPE binders showed perfect flexibility at the low temperature that the S-1, S-2, S-3, and S-4 binder systems obtained the values of 390.95 ± 39.716%, 1493.50 ± 47.681%, 1184.580 ± 95.276%, and 1371.1 ± 116.196%, respectively, and the ε_b_ also dropped with a steep bridge from −40 °C to 20 ℃. For the Young’s modulus, the S-1, S-2, and S-4 binder systems exhibited similar changes with the increasing temperature, and all obtained the maxima of 6.234 ± 0.246 GPa, 8.113 ± 0.238 GPa, 11.759 ± 0.214 GPa, and 2.736 ± 0.126 GPa at 40 °C. Inversely, the S-4 binder system obtained a maximum of 9.938 ± 0.13 GPa at the high temperature. The good mechanical properties were mainly determined by the HTPE polymers, and the differences appeared because the HTPE polymers produced different crosslinked polymeric matrices (polyurethanes) with different isocyanates. These results not only exactly verified the DMA analysis, but also confirmed the molecular dynamic results.

#### 4.2.4. Morphology

For the solid propellants, the solid segregation, the subsequent preferential wetting of the particles, the energy liquid migration, and the fractures of the binders could cause terrible detriments to the mechanical properties. So, it is important to understand the fractured mechanism of the binders and improve the mechanical properties of propellants. The SEM micrographs of the detailed fractured surface structures are shown in Figure 12a HTPE/N-100, (b) HTPE/HDI, (c) HTPE/TDI, and (d) HTPE/IPDI. All the HTPE binders showed homogeneous and dense polyurethane matrixes. For thermoplastic polyurethane elastomers, the fracture form of weak intermolecular force materials was basically the slip between molecular chains. The tensile fracture of materials with strong intermolecular forces not only [50] led to the slip of molecular chains, but also caused some of the main chains to fracture, so as to improve the tensile strength [51,52]. Due to the wide crosslinking structures and strong van der Waals interactions, many bundles and stripes emerged in the sights, and cracking propagated rips along the debonding surface. Moreover, the different tensile fractured HTPE binders revealed the various microcrack mechanisms. HTPE/N-100, HTPE/TDI, and HTPE/IPDI binders showed strong plastic deformation characteristics. However, the HTPE/HDI binders showed smoother cracks. These results provided evidence on how the microstructures of prepolymer crosslinked with different curing agents effected the mechanical properties and provided evidence to the front molecular dynamic simulations.

## 5. Conclusions

In this paper, the molecular dynamics simulations and experimental investigations were conducted to explore the mechanical properties of HTPE binder systems. The summaries are as follows:

The crosslinked models similar to the experimental ones could be gained automatically by Perl scripts. The glass transition temperatures of the S-1, S-2, S-3, and S-4 binder systems were 252.567 K, 244.573 K, 257.87 K, and 240.588 K, respectively. The mechanical properties of the HTPE binders showed that the S-2 binder system had the best mechanical parameters. The bond-length distributions of the prepolymers and the curing agents both were different from the crosslink structures. The conformation property analysis confirmed that the S-3 binding system had the highest synergetic rotational energy, 7979.79 kcal/mol, and the S-4 binding system had the lowest one, 6892.81 kcal/mol. The cohesive energy density verified that the S-4 binding system had the lowest, 3.062 × 10^8^ J/m^3^. The fraction free volume demonstrated that the S-2, S-3, and S-4 binder systems had similar FFVs, which were all lower than the S-1 binder system.

In terms of the molecular dynamic results, four curing specimens of HTPE binders were prepared and tested. The FTIR-ATR analysis explained that the crosslinked polyurethane structures had been synthesized between prepolymers and curing agents. The dynamic mechanical analysis indicated a closed T_g_ of approximately −68.0 °C, which proved that the prepolymers played a main role in the mechanical properties of HTPE binder systems. The mechanical properties of the uniaxial machine showed that the ultimate stresses and Young’s modulus of the HTPE binders decreased with rising temperatures, while the strains at break were in a growing trend. The SEM morphology of the fractured surface confirmed that the failure modes of the HTPE binders were mainly intermolecular slipping and molecular chain breakage.

From the analysis of the four curing crosslinking systems, molecular dynamic simulations predicted the mechanical behaviors and explained the thermomechanical mechanisms of the HTPE binders, which were in good agreement with experimental specimens.

## Figures and Tables

**Figure 1 polymers-14-05491-f001:**
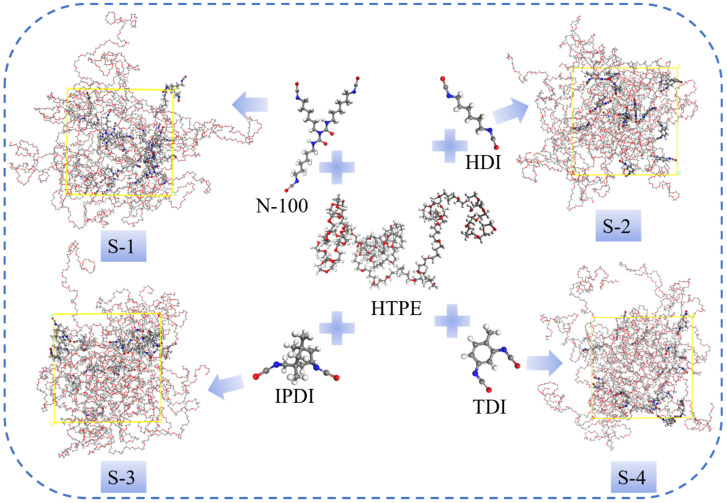
Molecules of prepolymer, curing agents, and blended models of HTPE binder systems.

**Figure 2 polymers-14-05491-f002:**
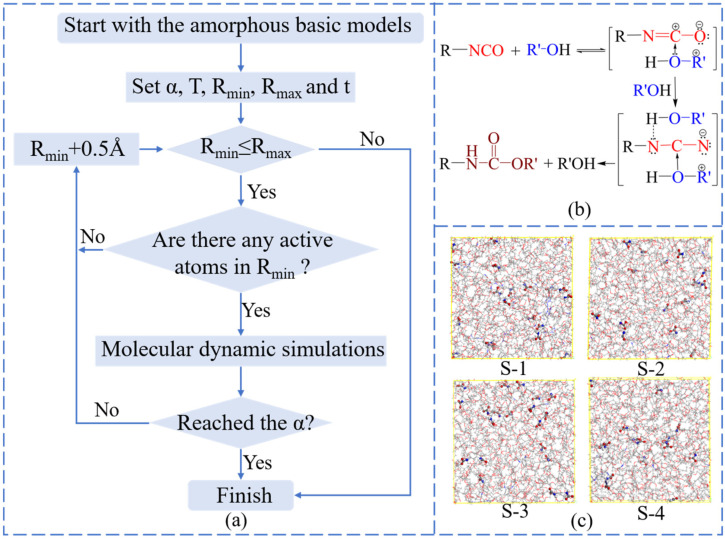
(**a**) The flow chart of the crosslinking reaction algorithm; (**b**) the curing reaction mechanism of HTPE binders; (**c**) the crosslinking basic models of HTPE binders.

**Figure 3 polymers-14-05491-f003:**
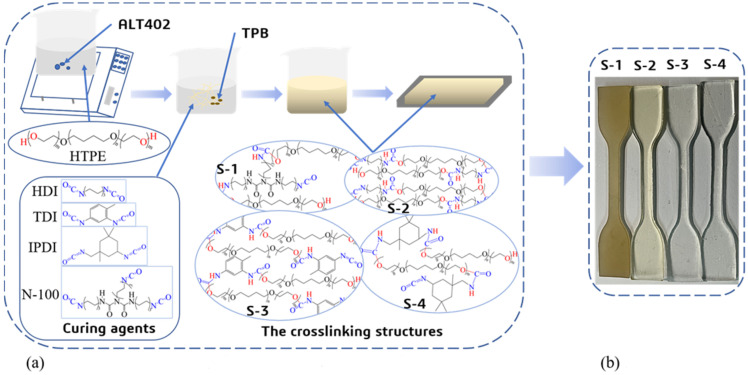
The simplified experimental routes to process the HTPE binders. (**a**) the diagram illustration of the preparation process of HTPE binders; (**b**) the HTPE binders.

**Figure 4 polymers-14-05491-f004:**
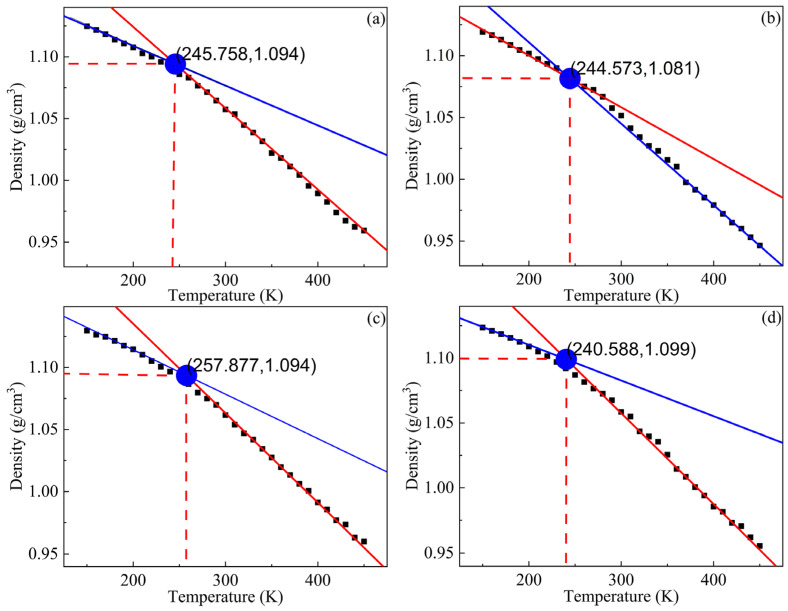
The glass transition temperatures of HTPE binder systems. (**a**) S-1, (**b**) S-2, (**c**) S-3 and (**d**) S-4.

**Figure 5 polymers-14-05491-f005:**
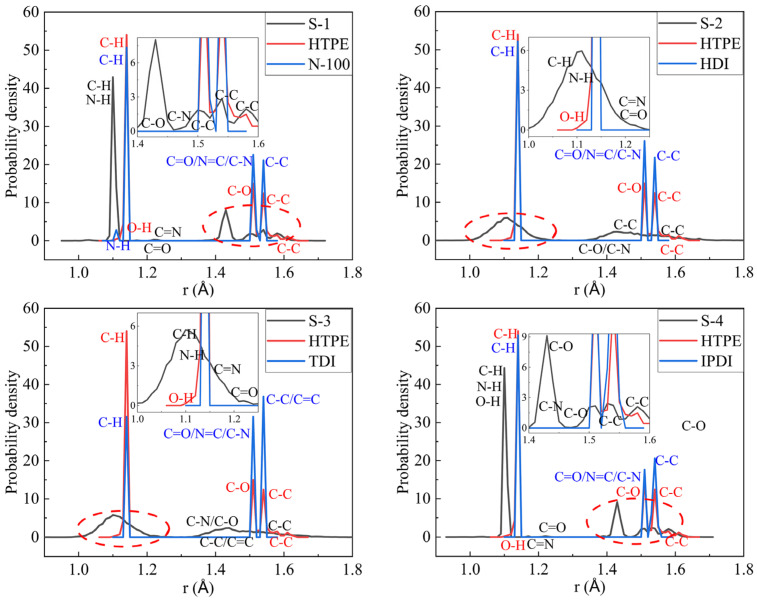
The bond-length distributions of HTPE binder systems.

**Figure 6 polymers-14-05491-f006:**
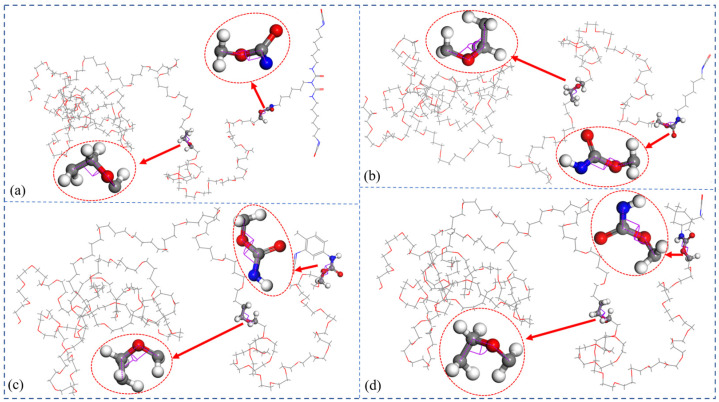
The torsion of rotation of two specific bonds Φ1 and Φ2 in four HTPE polyurethanebinders. (**a**) HTPE/N-100, (**b**) HTPE/HDI, (**c**) HTPE/TDI and (**d**) HTPE/IPDI.

**Figure 7 polymers-14-05491-f007:**
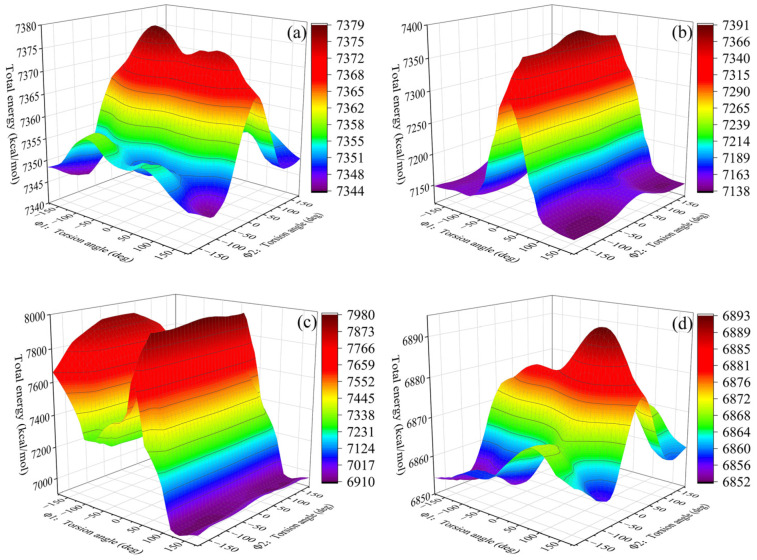
The torsional rotational of the specific bonds. (**a**) S-1, (**b**) S-2, (**c**) S-3 and (**d**) S-4.

**Figure 8 polymers-14-05491-f008:**
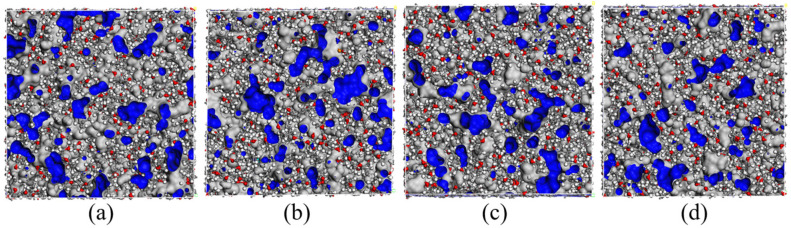
The free volumes of HTPE binders. (**a**) S-1, (**b**) S-2, (**c**) S-3 and (**d**) S-4.

**Figure 9 polymers-14-05491-f009:**
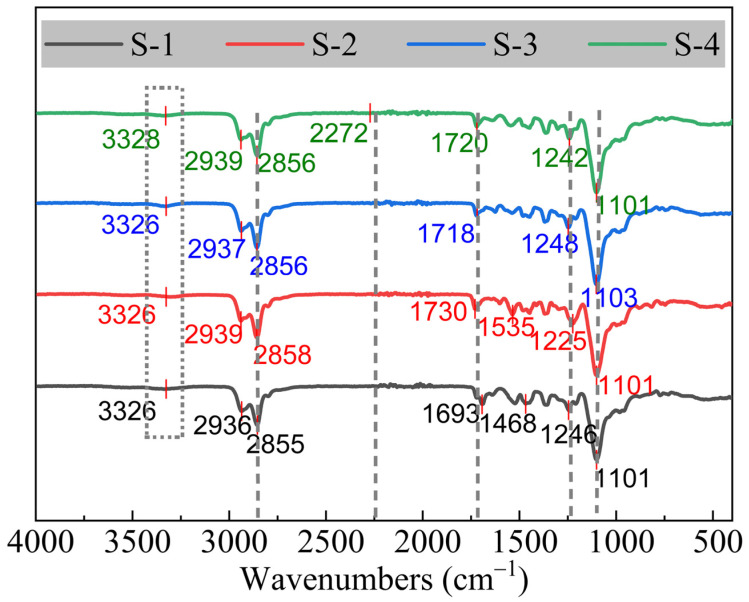
FTIR-ATR spectra of HTPE polyurethane binders.

**Figure 10 polymers-14-05491-f010:**
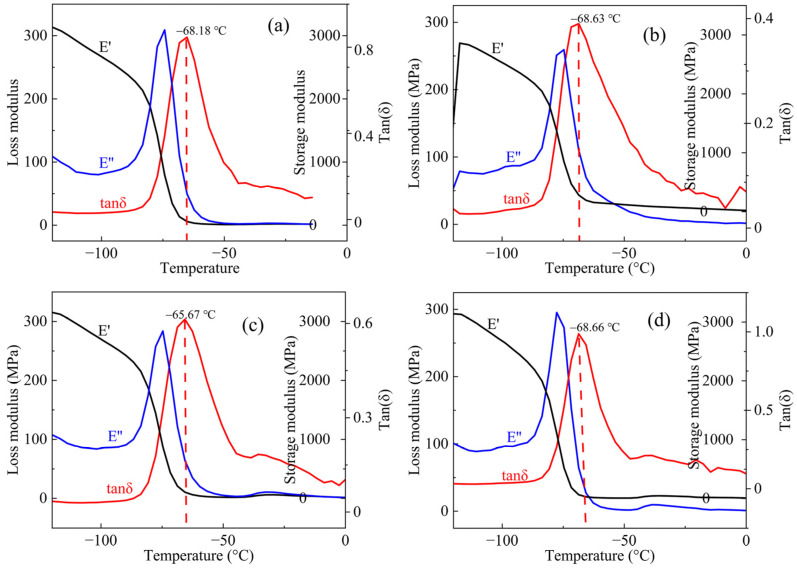
The DMA curves of four HTPE binder specimens. (**a**) S-1, (**b**) S-2, (**c**) S-3 and (**d**) S-4.

**Figure 11 polymers-14-05491-f011:**
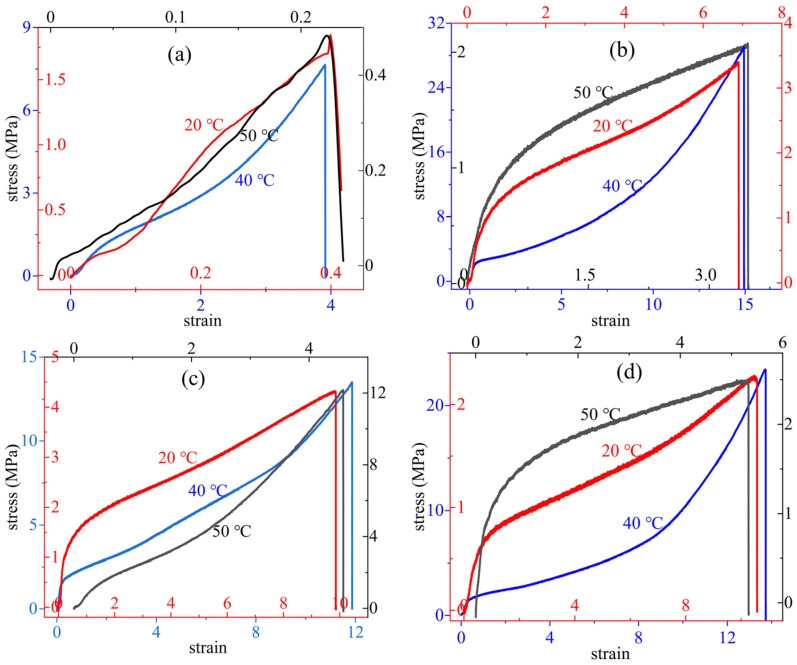
The strain-stress curves of four HTPE binder systems. (**a**) S-1, (**b**) S-2, (**c)** S-3 and (**d**) S-4.

**Figure 12 polymers-14-05491-f012:**
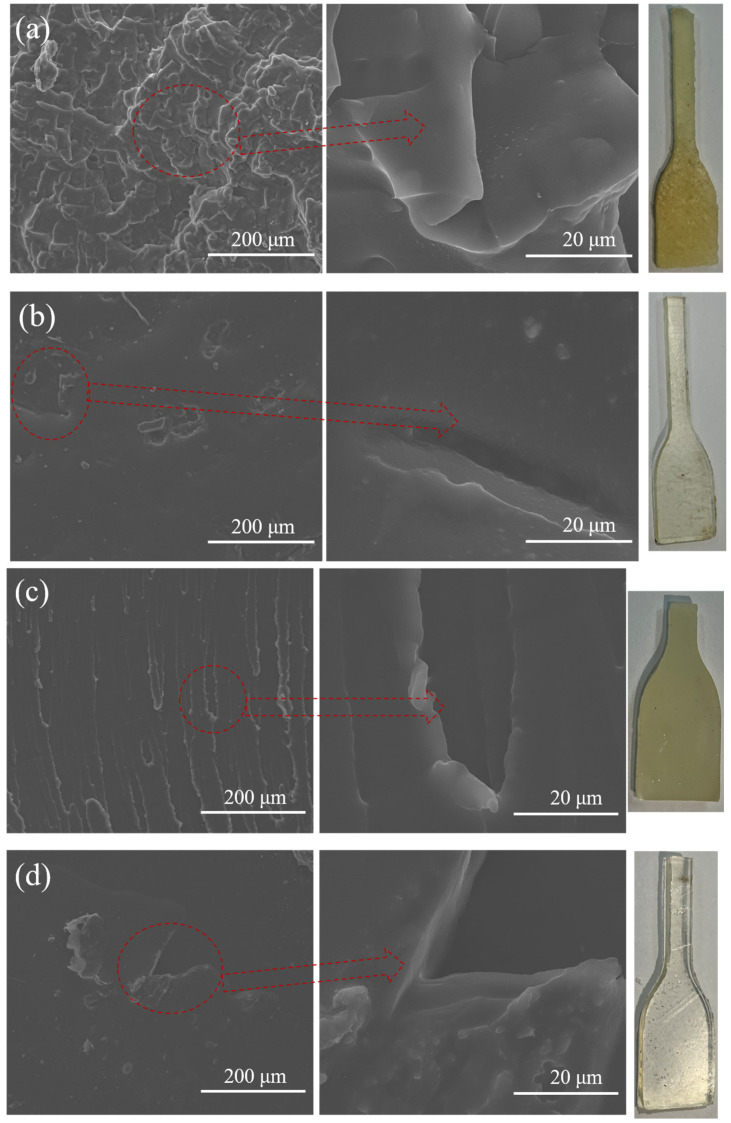
The SEM micrographs of the HTPE binders (**a**) HTPE/N-100, (**b**) HTPE/HDI, (**c**) HTPE/TDI, and (**d**) HTPE/IPDI.

**Table 1 polymers-14-05491-t001:** The simulated mechanical properties of S-1, S-2, S-3 and S-4 binders.

Systems	Bulk Modulus/GPa (K)	Shear Modulus/GPa (G)	Young’s Modulus/GPa (E)	Poisson’s Ratio (𝛖)
S-1	5.657	1.877	5.070	0.351
S-2	6.285	2.021	5.477	0.355
S-3	5.923	1.850	5.026	0.359
S-4	6.231	1.793	4.909	0.369

**Table 2 polymers-14-05491-t002:** The cohesive energy density of HTPE binders.

Systems	S-1	S-2	S-3	S-4
CED (J/m^3^)	4.579 × 10^8^	3.138 × 10^8^	3.139 × 10^8^	3.062 × 10^8^

**Table 3 polymers-14-05491-t003:** The results of V_0_, V_f_, and FFV in the HTPE binder systems.

Systems	V_0_ (Å^3^)	V_f_ (Å^3^)	FFV (%)
S-1	59,555.74	16,790.48	22.0
S-2	57,424.65	12,483.91	17.9
S-3	56,915.3	12,417.41	17.9
S-4	58,395.18	12,810.76	18.0

**Table 4 polymers-14-05491-t004:** The σ_m_, ε_b,_ and E_exp_ of the S-1, S-2, S-3, and S-4 binder systems.

System	Temperature/°C (T)	Stress/MPa/(σ_m_)	Strain/% (ε_b_)	Young Modulus/GPa (E_exp_)
S-1	−40	8.356 ± 1.949	390.95 ± 39.716	6.234 ± 0.246
	20	0.812 ± 0.350	25.453 ± 11.528	4.191 ± 0.162
	50	0.4668 ± 0.271	22.27 ± 10.314	2.714 ± 0.100
S-2	−40	29.057 ± 0.559	1493.50 ± 47.681	8.113 ± 0.238
	20	3.4031 ± 0.146	688.8 ± 36.595	6.367 ± 0.138
	50	2.075 ± 0.104	347.82 ± 35.544	2.393 ± 0.146
S-3	−40	13.405 ± 2.001	1184.580 ± 95.276	11.759 ± 0.214
	20	3.519 ± 0.476	695.483 ± 69.69	6.297 ± 0.145
	50	1.4642 ± 0.282	892.72 ± 39.41	5.218 ± 0.157
S-4	−40	24.435 ± 3.841	1371.1 ± 116.196	2.736 ± 0.126
	20	2.279 ± 0.445	1045.7 ± 196.194	1.284 ± 0.145
	50	2.4959 ± 0.326	510.78 ± 85.612	9.938 ± 0.13

## Data Availability

Data available on request due to privacy.

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
