# Peer review of "Molecular Dynamic Simulations and Experiments Study on the Mechanical Properties of HTPE Binders"

_polymers, 2022, doi:10.3390/polym14245491_

Round 1
Reviewer 1 Report
In this work, the crosslinking structures of HTPE binders were established by a computational procedure, and experiments provides some more evidences for the findings. The work is interesting but requires some modification: 1. The Abstract should be well written. 2. The recent literature works should be cited in the Introduction. 3. How to test the mechanical properties between - 40 and 50 degrees and How to guarantee the sample at such a temperature range. 4. Reference: The references should be as per journal style. 5. There are few grammatical errors which need to be taken care during revised submission.Author Response
Response to Reviewer Comments
Referee: 1
Comments:
In this work, the crosslinking structures of HTPE binders were established by a computational procedure, and experiments provides some more evidences for the findings. The work is interesting but requires some modification:
Point 1: The Abstract should be well written.
Response 1: Thank you for this suggestion and we have made some careful modifications in the Abstract section.
Point 2: The recent literature works should be cited in the Introduction.
Response 2: Thank you for this suggestion and we have added some recent literature works in the Introduction section. For example,
- Yuan, S.; Zhao, Y.; Luo, Y. Rheological properties of HTPE/PCL four-componenet propellant slurry. Chinese Journal of Explosives & Propellants 2021, 44, 7.
- Fu, X.-l.; Fan, X.-z.; Ju, X.-h.; Qi, X.-f.; Li, J.-z.; Yu, H.-j. Molecular dynamic simulations on the interaction between an HTPE polymer and energetic plasticizers in a solid propellant. Rsc Advances 2015, 5, 52844-52851.
- Fu, X.-l.; Fan, X.-z. Curing reaction kinetics of HTPE polymer studied by simultaneous rheometry and FTIR measurements. Journal of Thermal Analysis and Calorimetry 2016, 125, 977-982.
- Chen, K.; Yuan, S.; Wen, X.; Sang, C.; Luo, Y. Effect of Mixed Isocyanate Curing Agents on the Performance of In Situ‐Prepared HTPE Binder Applied in Propellant. Propellants, Explosives, Pyrotechnics 2021, 46, 428-439.
- Dong, M.; Li, Q.; Liu, H.; Liu, C.; Wujcik, E.K.; Shao, Q.; Ding, T.; Mai, X.; Shen, C.; Guo, Z. Thermoplastic polyurethane-carbon black nanocomposite coating: fabrication and solid particle erosion resistance. Polymer 2018, 158, 381-390.
- Guo, Y.; Zhang, R.; Xiao, Q.; Guo, H.; Wang, Z.; Li, X.; Chen, J.; Zhu, J. Asynchronous fracture of hierarchical microstructures in hard domain of thermoplastic polyurethane elastomer: Effect of chain extender. Polymer 2018, 138, 242-254.
- Pattanayak, A.; Jana, S.C. Properties of bulk-polymerized thermoplastic polyurethane nanocomposites. Polymer 2005, 46, 3394-3406.
Point 3: How to test the mechanical properties between - 40 and 50 degrees and How to guarantee the sample at such a temperature range.
Response 3: Thank you for this suggestion. For this uniaxial tensile in special temperatures, the universal testing machine (AG-X plus, 5kN, shimadzu, Japan) was equipped with an incubator, which could guarantee the sample testing at special temperatures. Before the test, the specimens would keep at the special temperature for two hours to make the whole specimen thermal equilibrium. Then the tensile test was carried out in the same closed incubator until it broke.
Point 4: Reference: The references should be as per journal style.
Response 4: Thank you for this suggestion and we have modified the references as MDPI styles in the Reference section.
Point 5: There are few grammatical errors which need to be taken care during revised submission.
Response 5: Thank you for this suggestion. And we have carefully revised the whole text as the attachment.

Reviewer 2 Report
This manuscript is relevant and may benefit from important improvements.
1 - The authors are encouraged to seek the aid from a native English speaker to revise the text. Several language flaws are really an handicap, compromising the clearness of the presentation. For example, sections titles 3.3 Characteristic and 4.2.4 The Morphology Characteristic should be appropriately renamed.
2 - In Figure 2b) the yellow molecular structure is hardly seen. Try change its colour.
3 - In these molecular dynamics simulations, using Materials Studio, which were the approximations used and which were the wavefunctions bases chosen for the different steps of the process? For quantum molecular dynamics simulations these two pieces of information are really mandatory.
Author Response
Response to Reviewer Comments
Comments:
This manuscript is relevant and may benefit from important improvements.
Point 1: The authors are encouraged to seek the aid from a native English speaker to revise the text. Several language flaws are really an handicap, compromising the clearness of the presentation. For example, sections titles 3.3 Characteristic and 4.2.4 The Morphology Characteristic should be appropriately renamed.
Response 1: Thank you for these your perfect suggestions and we have revised the whole text as attachment. Especially, for some section titles, 3.3 Characteristic was changed into Apparatus and characterizations; 4.2.4 The Morphology Characteristic was changed into Morphology;
Point 2: In Figure 2b) the yellow molecular structure is hardly seen. Try change its colour.
Response 2: Thank you for this detail and we have revised it as attachments.
Point 3: In these molecular dynamics simulations, using Materials Studio, which were the approximations used and which were the wavefunctions bases chosen for the different steps of the process? For quantum molecular dynamics simulations these two pieces of information are really mandatory.
Response 3: Thank you for this question and here were some ideas that I have looked for some literatures. Accurate simulation of atomic and molecular systems generally involves the application of quantum mechanical theory. However, quantum mechanical techniques are computationally expensive and are usually only applied to small systems containing between 10 and 100 atoms, or small molecules. It is not monomers in this molecular dynamic simulations. Even if such a simulation were possible, in many cases much of the information generated would be discarded. This is because in simulating large systems, the goal is often to extract bulk (statistical) properties, such as diffusion coefficients or Young’s moduli, which depend on the location of the atomic nuclei or, more often, an average over a set of atomic nuclei configurations.
Molecular mechanics was also called force field method to calculate the equilibrium structure and energy of molecules based on the classical Newtonian equations of mechanics. Unlike quantum mechanics, it solves Newton’s equation rather than Schrodinger’s.
Under these circumstances the details of electronic motion are lost in the averaging process, so bulk properties can be extracted if a good approximation of the potential in which atomic nuclei move is available and if there are methods that can generate a set of system configurations which, while they may not follow the exact dynamics of the nuclei, are statistically consistent with a full quantum mechanical description.
Some based assumptions: The Born-Oppenheimer Approximation; Simple action models; Force filed portability.
Molecular force field is the core of molecular mechanics. The basic theory of molecular mechanics is that a molecular force field consists of two parts: intramolecular interaction and intermolecular interaction, that is, the potential energy of a force field includes bonding and non-bonding interactions, and the sum of all potential energy is the imagined energy of a molecule.
In this paper, compass Ⅱforce filed and forcefield assigned charges were used in all molecular dynamic simulations. And the calculation was fine (15.5 Å). For the summation method of blend models, the electrostatic was Ewald and van der Waals was group based. For the single molecular chain, both the electrostatic and van der Waals were group based.
The purpose of a molecular dynamics simulation is to integrate Newton’s equations of motion for the set of particles in the given system. It approximates the time evolution of the equations of motion of the system.

Round 2
Reviewer 2 Report
Relative to the issues addressed in the first review, the authors properly replied to point 1.
As to point 2, Figure 2 remains unchanged, although they have improved figure 6.
As to point 3, the explanation provided is rather pertinent, however it is absent in the revised manuscript. It should be considered to incorporate the most important aspects of those comments in section 2.1 The models construction and Molecular Dynamics simulations.
A file with Figure 2 and some highlights of the MD method is attached to assist the revision.

Author Response
Point 2: In Figure 2b) the yellow molecular structure is hardly seen. Try change its colour.
Response 2: Thank you for this detail and we have revised it as follows, which also could be seen in the revised manuscript.
As to point 3, the explanation provided is rather pertinent, however it is absent in the revised manuscript. It should be considered to incorporate the most important aspects of those comments in section 2.1 The models construction and Molecular Dynamics simulations.
Response 3: Thank you for this detail and we have revised it as follows, which also could be seen in the revised manuscript.
2.1 The model constructions and molecular dynamic simulations
The all molecular models were built by Materials Studio and the constructions could refer to the literatures[35,36]. The structures of Hydroxy-terminated polyether (HTPE), polyfunctional isocyanate (N-100), hexamethylene diisocyanate (HDI), toluene diisocyanate (TDI), isophorone diisocyanate (IPDI) and the four blends were shown in Figure 1. The HTPE chains were built with repeat ethylene glycol and tetrahydrofuran in the Builder Polymers module, which average molar weight was 3974g/mol.
The blends were constructed in the Amorphous Cell module with the box 90.39 Å×89.46 Å×90.48 Å and the initial density was 0.6 g/cm3. Then 10 frames would be output and the lowest energy one was expected to the basic structure. The van der Waals and electrostatic interactions selected the Atom-based and Ewald method. Meanwhile, the simulation quality would be Fine.
Molecular force field was the core of molecular mechanics, which calculated the equilibrium structure and energy of molecules based on the classical Newtonian equations of mechanics. Bulk properties of HTPE binder models could be extracted if a good approximation of the potential in which atomic nuclei move was available and if there methods that could generate a set of system configurations which, were statistically consistent with a full quantum mechanical description. After the molecular models were built, the geometry structures would be relaxed with 50000 steps energy minimization. Simulations started by choosing an initial state by setting the initial positions and velocities of all the particles. The initial velocities were usually set by choosing each component of the velocity vetor of each particle from the Maxwell distribution. HTPE/N-100, HTPE/HDI, HTPE/TDI and HTPE/IPDI binder models were simplified as S-1, S-2, S-3 and S-4, respectively. Then the amorphous blends were annealed from 600 K to 300 K with the interval temperature 20 K and 10 annealing recycles in total at each temperature, which contained 200 picosecond (ps) dynamics performing every temperature ramp in the NPT ensemble, NHL thermostat and Berendsen [37] barostat. Furthermore, the time step was 1.0 femtosecond, which was chosen to be small as compared to be the shortest fundamental time scale in the Hamiltonian, while not so small as limit the efficiency of the program. The MD time average were equivalent to averages over the microcanonical ensemble. These models need further dynamics to equilibrate the system with the NPT [38] ensemble for 400ps. In this paper, all models were optimized and analyzed under CompassⅡ force filed [39] and forcefield assigned charges.
